# The kinetic profiles of copeptin and mid regional proadrenomedullin (MR-proADM) in pediatric lower respiratory tract infections

Philipp Baumann[1¤a]*, Aline Fuchs[2], Verena Gotta[2], Nicole Ritz[1,2], Gurli Baer[1], Jessica M. Bonhoeffer[3¤b], Michael Buettcher[4¤c], Ulrich Heininger[1], Gabor Szinnai[5,6☯], Jan Bonhoeffer[1☯], for the ProPAED study group[¶]

1 Department of Paediatric Infectious Diseases and Vaccines, University of Basel Children's Hospital, Basel, Switzerland, 2 Department of Paediatric Pharmacology and Pharmacometrics, University of Basel Children's Hospital, Basel, Switzerland, 3 Department of Paediatrics, University of Basel Children's Hospital, Basel, Switzerland, 4 Department of Paediatrics, Kantonsspital Aarau, Aarau, Switzerland, 5 Department of Paediatric Endocrinology and Diabetology, University of Basel Children's Hospital, Basel, Switzerland, 6 Department of Clinical Research, University Hospital Basel, University of Basel, Basel, Switzerland

☯ These authors contributed equally to this work.
¤a Current address: Department of Intensive Care Medicine and Neonatology, University Children's Hospital Zurich, Zurich, Switzerland
¤b Current address: Child Development Center, University Children's Hospital Zurich, Zurich, Switzerland
¤c Current address: Department of Paediatric Infectious Diseases, Children's Hospital Lucerne, Cantonal Hospital Lucerne, Lucerne, Switzerland
¶ Acknowledgments section provides all members of the ProPAED study group.
* philipp.baumann@kispi.uzh.ch

**Data Availability Statement:** All relevant data are within the paper and its supporting information files.

## Abstract

### Background

Kinetics of copeptin and mid regional proadrenomedullin (MR-proADM) during febrile pediatric lower respiratory tract infections (LRTI) are unknown. We aimed to analyze kinetic profiles of copeptin and MR-proADM and the impact of clinical and laboratory factors on those biomarkers.

### Methods

This is a retrospective post-hoc analysis of a randomized controlled trial, evaluating procalcitonin guidance for antibiotic treatment of LRTI (ProPAED-study). In 175 pediatric patients presenting to the emergency department plasma copeptin and MR-proADM concentrations were determined on day 1, 3, and 5. Their association with clinical characteristics and other inflammatory biomarkers were tested by non-linear mixed effect modelling.

### Results

Median copeptin and MR-proADM values were elevated on day 1 and decreased during on day 3 and 5 (-26%; -34%, respectively). The initial concentrations of MR-proADM at inclusion were higher in patients receiving antibiotics intravenously compared to oral administration (difference 0.62 pmol/L, 95%CI 0.44;1.42, p<0.001). Intensive care unit (ICU) admission was associated with a daily increase of MR-proADM (increase/day 1.03 pmol/L,

**Funding:** Copeptin proAVP and MR-proADM test kits were provided by B.R.A.H.M.S. The funders had no role in study design, data collection and analysis, decision to publish, or preparation of the manuscript.

**Competing interests:** Copeptin proAVP and MR-proADM test kits were provided by B.R.A.H.M.S. There are no patents, products in development or marketed products associated with this research to declare. This does not alter our adherence to PLOS ONE policies on sharing data and materials.

95%CI 0.43;1.50, p<0.001). Positive blood culture in patients with antibiotic treatment and negative results on nasopharyngeal aspirates, or negative blood culture were associated with a decreasing MR-proADM (decrease/day -0.85 pmol/L, 95%CI -0.45;-1.44), p<0.001).

## Conclusion

Elevated MR-proADM and increases thereof were associated with ICU admission suggesting the potential as a prognostic factor for severe pediatric LRTI. MR-proADM might only bear limited value for decision making on stopping antibiotics due to its slow decrease. Copeptin had no added value in our setting.

## Introduction

Lower respiratory tract infection (LRTI) is a leading cause of morbidity and mortality in children and adolescents worldwide [1, 2]. It may lead to severe complications like pleural effusions, empyema or necrotizing pneumonia [3, 4]. While some patients with LRTI benefit from antibiotic treatment, it is estimated that up to 90% are unnecessarily treated [5–7]. Biomarkers could help to rule-out LRTI requiring antibiotic treatment [8–10], but sensitivity and positive predictive value for ruling-in the need for antibiotic treatment are still low [11–13]. Ongoing research on new biomarkers with better test accuracy has generated promising results for two candidate markers, copeptin (CT-proAVP) and mid regional proadrenomedullin (MR-proADM). Copeptin is a 39-amino acid glycopeptide representing the C-terminal part of the vasopressin precursor molecule pre-provasopressin. It is secreted at equimolar concentrations from the neurohypophysis together with vasopressin and neurophysin II upon stimulation [14]. First studies on copeptin in pediatric LRTI demonstrated that copeptin might be used to diagnose community-acquired pneumonia (CAP) and to predict development of complications, but the results of these studies were contradictory [15–19] and kinetics over time have not been assessed yet. MR-proADM is a 48-amino acid fragment split from the adrenomedullin precursor pre-pro-adrenomedullin in a 1:1 ratio. MR-proADM reflects the adrenomedullin translation and was studied in pediatric LRTI in recent years in the context of assessing infectious disease severity [18, 20, 21]. Kinetic profiles on MR-proADM have been found to add prognostic value for predicting adverse outcome in adult LRTI [22], but have not been evaluated in children so far. Therefore, we aimed at (i) determining the kinetic profiles of copeptin and MR-proADM over five days in children and adolescents admitted to the emergency department for suspected LRTI, and (ii) investigating the influence of clinical and laboratory covariates as well as the development of LRTI complications on copeptin and MR-proADM course over time in a retrospective subanalysis of the ProPAED trial [9].

## Methods

### Study design and population

This study is a protocol-specified post-hoc analysis of a randomized controlled trial, evaluating PCT as a biomarker guiding antibiotic treatment of LRTI in children and adolescents [9]. Patients (1 month– 18 years) presenting with febrile LRTI to emergency departments of two Swiss pediatric hospitals (Basel and Aarau) between January 2009 and February 2010 were eligible. LRTI was defined by the presence of fever (≥38˚C measured in the hospital or at home), and at least one of the following symptoms: tachypnea, dyspnea, wheezing, late inspiratory

crackles, bronchial breathing, and/or pleural rub. Patients with severe primary or secondary immunodeficiency, immunosuppressive treatment, neutropenia (<1 G/L), cystic fibrosis, upper respiratory tract infection or hospital stay within previous 14 days were excluded. In the ProPAED trial patients were randomized to a PCT guided intervention group or to a standard care control group with treatment based on international guidelines [9, 23].

## Variables

The following variables were prospectively recorded at inclusion or prior to these ancillary analyses as appropriate: age, days of fever before presentation, sex, antibiotic treatment, antibiotic pre-treatment before study inclusion, diagnosis, complications (parapneumonic effusion, empyema, acute respiratory distress syndrome (ARDS), sepsis, shock), body temperature, heart rate, breath rate, oxygen administration, hospitalization, admission to intensive care unit (ICU). C-reactive protein (CRP), PCT and cytokine (interferon (IFN)-γ, interleukin (IL)-1ra, IL-1β, IL-2, IL-4, IL-6, IL-10, IFN-γ-inducible protein (IP)-10 and tumor necrosis factor (TNF)-α) concentrations were recorded prospectively on day 1, 3 and 5 after inclusion [9, 24]. Blood cultures (BC) and nasopharyngeal aspirates (NPA) were performed at physician's discretion. Microbiology results were classified in 4 categories: not performed, negative, systemic bacterial infection (blood culture positive for pneumococcus/streptococcus = invasive pneumococcal infection) and other. Copeptin and MR-proADM were measured for the present analysis, if the stored EDTA plasma sample volumes remaining from the ProPAED trial were large enough for three post-hoc biomarker measurements (day 1–3–5). Heart rate was classified as normal or elevated according to age-dependent references ranges, respiratory rate was classified as elevated if exceeding WHO age-dependent reference ranges for children up to 5 years of age, and if exceeding percentile 90 as reported by Fleming et al. for children >5 years of age (S1 Table) [25, 26].

## Assays

EDTA plasma collected on day 1, 3 and 5 after inclusion was stored at– 80˚C. Copeptin and MR-proADM levels were measured using the commercial sandwich immunoluminometric assays B·R·A·H·M·S™ Copeptin proAVP KRYPTOR™ and B·R·A·H·M·S™ MR-proADM KRYPTOR™ (B·R·A·H·M·S GmbH, part of Thermo Fisher Scientific, Henningsdorf, Germany). The lower limit of quantitation of the assays was 1.23 pmol/L and 0.23 nmol/L, the upper limit of quantitation was 2000 pmol/L and 100 nmol/L for copeptin and MR-proADM, respectively. For MR-proADM, the manufacturer reported for the values between 0.2–0.5 nmol/L and between 0.5–6 nmol/L intra assay precisions of ≤10% and <2–4% and inter assay precisions of ≤18% and <6–11%, respectively. For copeptin, the manufacturer reported for the values between 2–4 pmol/L and between 4–50 pmol/L intra assay precisions of ≤15% and <4–8% and inter assay precisions of ≤18% and <6–10%, respectively. According to maximum inter-assay coefficient of variation (CV%), an increase was set as significant of at least 18% and 20% for copeptin and MR-proADM concentrations, respectively.

## Statistical analysis

Population characteristics and clinical findings were described by standard descriptive statistics (median and interquartile range (IQR) for continuous variables, count and proportion for categorical variables).

**Non-linear mixed effect modelling.** Copeptin and MR-proADM kinetics and their association with clinical parameters, laboratory variables, patient management, microbiology, and chest radiography were first investigated by visual inspection. Characterization of copeptin

and MR-proADM kinetics, and quantification of inter- and intra-individual variability was performed using a nonlinear mixed effect modeling approach with NONMEM® (version 7.1.3, ICON Development Solutions, Ellicott City, MD, USA). Fitting was performed for each biomarker (copeptin and MR-proADM) separately. Model estimation was performed using the first-order conditional estimation with interaction (FOCE-I / LAPACE). Details on non-linear effect modeling, parameter estimation, model selection and model evaluation are available in S1 Appendix.

**Influence of patient characteristics.** Association with clinical parameters, laboratory variables, patient management, microbiology and, chest radiography, called covariates, were investigated on model parameters on which variability was identified. The stepwise covariate model building approach is explained in S1 Appendix.

**Correlations between biomarkers.** The correlation between copeptin and MR-pro-ADM concentrations on patients'admission was investigated by Pearson correlation test to evaluate the strength of relationship. The correlation between copeptin, MR-proADM, C-reactive protein (CRP), procalcitonin (PCT), and various cytokines (interferon (IFN)-γ, interleukin (IL)-1ra, IL-1β, IL-2, IL-4, IL-6, IL-10, IFN-γ-inducible protein (IP)-10, and tumor necrosis factor (TNF)-α) was investigated on log-transformed variables. Graphical inspection, descriptive statistics and Pearson correlation test were performed with R software (version 3.1.2; R Development Core Team, Vienna, Austria, http://www.r-project.org).

## Ethics

Both pertinent ethics committees of the University of Basel and Kanton Aargau (Switzerland) approved the trial protocol. Written informed consent was obtained from all patients or their care takers. The trial was registered with the International Standard Randomized Controlled Trial Number Register (ISRCTN 17057980).

## Results

### Study population and clinical characteristics

The study population comprised 175 febrile LRTI pediatric patients (age: 1 month-18 years) for whom a sufficient quantity of blood plasma was in storage for copeptin and MR-proADM analysis for day 1, 3, and 5 after study inclusion. Blood sampling was performed in-house while patients were hospitalized or in the emergency department, to where the outpatients and the discharged patients were asked to return on day 3 and 5. Patients' baseline characteristics are summarized in Table 1.

### Kinetics of copeptin and MR-proADM concentrations

**Data exploration.** Median biomarker concentrations decreased over the 5 study days (Fig 1, for absolute biomarker values S4 Table). Age-stratification of copeptin and MR-proADM did not reveal any differences (S1 Fig and S2 Table). One patient (<1%) had increasing MR-proADM but not increasing copeptin concentrations and 11 patients (6%) had increasing copeptin concentrations over the 5 study days.

**Influence of patient characteristics on copeptin kinetics.** The search for factors influencing the course of copeptin over time in the covariate model did not reveal any parameter to have a significant impact on copeptin plasma concentrations over time or on its kinetics. For details see S1 Appendix.

**Influence of patient characteristics on MR-proADM kinetics.** During forward covariate model building, antibiotic administration route, general complication, microbiology results

**Table 1. Characteristics of the population included for present analyses and the ProPAED study population.**

| | Present analysis (n = 175) | ProPAED study (n = 337) |
|---|---|---|
| **Demographic** | | |
| Age, Median (IQR), years | 4.1 (1.9–6.6) | 2.8 (1.2–5.3) |
| *<1 yr, n (%)* | *16 (9)* | *71 (21)* |
| *1–5 yr, n (%)* | *87 (50)* | *171 (51)* |
| *5–18 yr, n (%)* | *72 (41)* | *95 (28)* |
| Male, n (%), gender | 98 (56) | 196 (58) |
| **Clinical features at inclusion (day 1)** | | |
| Antibiotic pre-treatment, n (%) | 26 (15) | 42 (12) |
| Days of fever before presentation, median (IQR) | 3 (1–4) | 3 (1–4) |
| Fever, n (%) | 175 (100) | 337 (100) |
| *Day 1, n (%)* | *132 (75)* | *247 (73)* |
| *Day 3, n (%)* | *14 (8)* | *34 (10)* |
| *Day 5, n (%)* | *4 (2)* | *11 (3)* |
| Fever before inclusion >1 day, n (%) | 122 (70) | 230 (68) |
| Cough, n (%) | 174 (99) | 336 (99) |
| Sputum production, n (%) | 73 (42) | 141 (42) |
| Poor feeding, n (%) | 63 (36) | 153 (45) |
| Pleuritic pain, n (%) | 58 (33) | 95 (28) |
| **Clinical findings at inclusion (day 1)** | | |
| Body temperature, median (IQR), °C | 38.5 (38–39.1) | 38.4 (37.9–39.1) |
| Respiratory rate, median (IQR), /min | 38 (28–44) | 40.0 (28.0–48.0) |
| Elevated breath rate* | | |
| *Day 1, n (%)* | *52 (30)* | *103 (31)* |
| *Day 3, n (%)* | *5 (3)* | *21 (6)* |
| *Day 5, n (%)* | *8 (5)* | *16 (5)* |
| Heart rate, median (IQR), b/min | 136.0 (120.0–159.0) | 142 (123–160) |
| *Day 1, n (%)* | *143 (82)* | *269 (80)* |
| *Day 3, n (%)* | *70 (40)* | *120 (36)* |
| *Day 5, n (%)* | *66 (38)* | *103 (31)* |
| Dyspnea, n (%) | 96 (55) | 217 (64) |
| Wheezing, n (%) | 47 (27) | 101 (30) |
| Late inspiratory crackles, n (%) | 62 (35) | 140 (41) |
| Reduced breathing sounds, n (%) | 60 (34) | 109 (32) |
| **Laboratory findings at inclusion (day 1)** | | |
| Procalcitonin, median (IQR), µg/L | 0.28 (0.13–2.50) | 0.24 (0.13–1.48) |
| Procalcitonin> 1 µg/L, n (%) | 61 (35) | 99 (29) |
| C-reactive protein, median (IQR) mg/L | 26.05 (10.25–93.38) | 20 (8–65) |
| C-reactive protein > 100 mg/L, n (%) | 40 (23) | 67 (20) |
| **Antibiotic prescription and admission** | | |
| Antibiotics within 14 days following randomization, n (%) | 105 (60) | 197 (58) |
| Antibiotics (AB) | | |
| *iv, n (%)* | *50 (29)* | *83 (25)* |
| *oral, n (%)* | *55 (31)* | *114 (34)* |
| *no AB, n (%)* | *70 (40)* | *140 (41)* |
| Hospitalization, n (%) | 89 (51) | 204 (60) |
| Admission to intensive care unit, n (%) | 4 (3) | 9 (3) |
| Supplemental oxygen, n (%) | 22 (12) | 79 (23) |

(*Continued*)

**Table 1.** (Continued)

| | Present analysis (n = 175) | ProPAED study (n = 337) |
|---|---|---|
| Complications[†], n (%) | 5 (3) | 6 (2) |
| **Microbiology** | | |
| Nasopharyngeal aspirate (NPA), n (%) | 168 (96) | 318 (94) |
| *M. pneumoniae or C. pneumonia, n (%)* | 13 (8) | 15 (4) |
| *human Metapneumovirus, n (%)* | 17 (10) | 42 (12) |
| *Influenza virus, n (%)* | 20 (12) | 36 (11) |
| *Respiratory syncytial virus, n (%)* | 26 (15) | 74 (22) |
| Blood culture, n (%) | 87 (50) | 148 (44) |
| *positive for S. pneumonia or S. pyogenes, n (%)* | 5 (3) | 6(2) |
| *no blood culture and NPA negative, n (%)* | 38 (21) | 69 (20) |
| *blood culture negative and NPA negative* | 16 (9) | 32 (9) |
| **Diagnosis** | | |
| Pneumonia | | |
| *Day 1, n (%)* | 82 (47) | 129 (38) |
| *Day 3, n (%)* | 84 (48) | 137 (41) |
| *Day 5, n (%)* | 84 (48) | 135 (40) |
| Bronchitis/-iolitis | | |
| *Day 1, n (%)* | 54 (31) | 122 (36) |
| *Day 3, n (%)* | 79 (45) | 148 (44) |
| *Day 5, n (%)* | 82 (47) | 165 (49) |
| Bronchitis/-iolitis + Pneumonia | | |
| *Day 1, n (%)* | 39 (22) | 86 (26) |
| *Day 3, n (%)* | 12 (7) | 48 (14) |
| *Day 5, n (%)* | 9 (5) | 31 (9) |

IQR = interquartile range.

[*] For age specific reference values for breath and heart rate see S1 Table.

[†]Complications were defined as sepsis, shock, pleural effusion, pleural empyema or acute respiratory distress syndrome (ARDS). Further details on age-stratified distribution of copeptin and MR-proADM in S1 Fig and in S2 Table. Five patients developed complications of LRTI. All recovered well, there were no deaths (S3 Table).

and presence of fever had a significant influence on MR-proADM concentrations on day 1 (p<0.001). Admission to ICU was associated to an increase of MR-proADM and microbiology results were associated with a decrease in MR-proADM. During the backward process, association with fever became not significant and was thus excluded. Details on base and final model can be seen in S1 Appendix. In the final model, intravenous administration of antibiotics influenced MR-proADM concentration and was significantly associated with a higher MR-proADM concentration on day 1. Further, ICU admission was significantly associated with an increased MR-proADM concentration. In contrast, positive blood cultures, which were grouped together with negative NPA or no growth in blood culture because of the same effect when building the model, led to a significant decrease of MR-proADM (Table 2).

## Correlation between copeptin and MR-proADM concentrations and pro- and anti-inflammatory markers

There was only a moderate positive correlation between copeptin and MR-proADM concentrations with inflammatory biomarkers, most noteworthy being IL-6 (r = 0.42 and 0.47

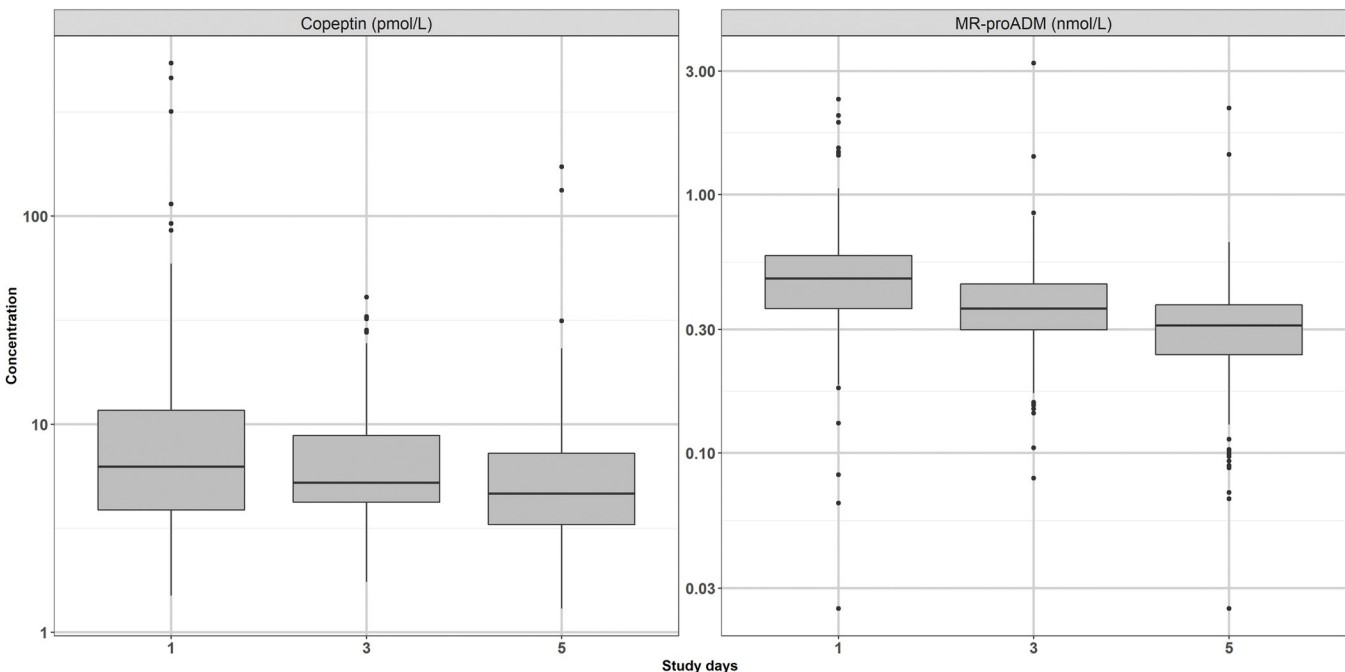

**Fig 1. Copeptin and MR-proADM concentrations over the study period.** Distribution and change in copeptin (pmol/L) and MR-proADM (nmol/L) concentrations for patients over 5 study days. Boxes represent the interquartile range (IQR). Solid lines are the median, 25th and 75th quantile and whiskers equal 25th quantile -1.5 IQR and 75th quantile +1.5 IQR.

respectively). MR-proADM and copeptin concentrations were not correlated to each other (r = 0.18). All correlations between copeptin, MR-proADM, CRP, PCT and various cytokines (interferon (IFN)-γ, interleukin (IL)-1ra, IL-1β, IL-2, IL-4, IL-6, IL-10, IFN-γ-inducible protein (IP)-10 and tumor necrosis factor (TNF)-α) on first admission to the emergency department (day 1) are presented in Fig 2.

## Discussion

This is the first study that investigated the kinetics of copeptin and MR-proADM in pediatric LRTI. Our main aim was to determine the dynamics of these biomarkers over the study period of five days. Our additional aim was to test for impact of clinical and laboratory markers that might indicate severity of disease on the biomarkers kinetics. Our results suggest, that MR-proADM but not copeptin might be helpful in differentiating severe from not severe pediatric LRTI cases.

In the recent past biomarker guided antibiotic therapy of LRTI was one research focus in infectious disease [9, 27]. Copeptin and MR-proADM have shown to discriminate between mild and severe LRTI in children if measured on first presentation to an emergency

**Table 2. MR-proADM: Variables effect on MR-proADM kinetics in multivariable analysis as compared to the typical patient.**

| | Definition of variable effect | Multivariable effect [95%CI] | p-value |
|---|---|---|---|
| MR-proADM concentration on day 1 | Administration of IV antibiotics | + 61% [39, 90] | <0.001 |
| MR-proADM decrease over 5 days | ICU-admission | + 107% [43, 150] | <0.001 |
| | Positive blood culture, negative NPA or no growth in blood culture | - 85% [−45, −144] | <0.001 |

IV: intravenous; ICU: intensive care unit; NPA: nasopharyngeal aspirates.

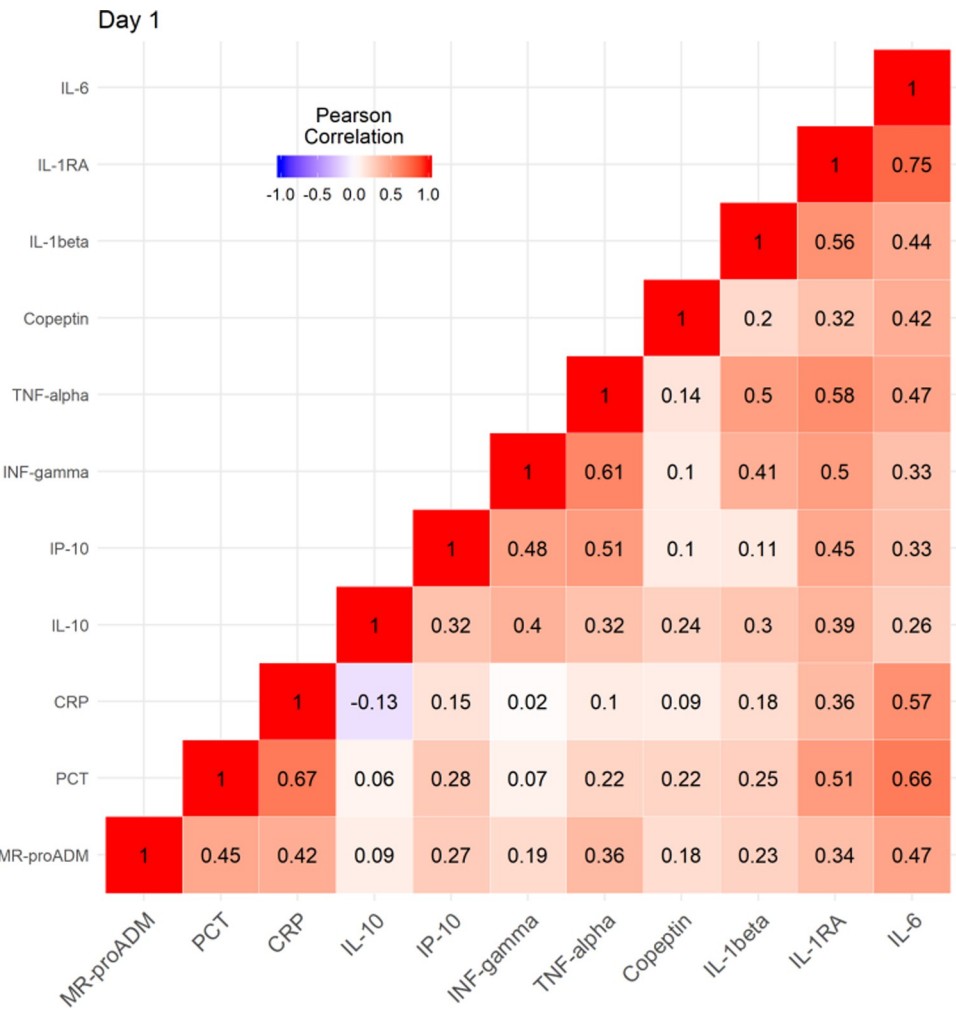

**Fig 2. Correlation between copeptin and MR-proADM concentrations and pro-and anti-inflammatory markers at study inclusion.** IL: interleukin; TNF: tumor necrosis factor; INF: interferon; IP-10: interferon-gamma induced protein 10 kD, CRP: c-reactive protein; PCT: procalcitonin; MR-proADM: mid regional proadrenomedullin.

department [16, 20]. Assuming that children with severe LRTI are more likely to have bacterial infection in need of antibiotic treatment, copeptin and MR-proADM could be used to start or withhold empiric antibiotic treatment. For monitoring during the course of disease and for the indication to stop antibiotics secondarily the course of these markers has to be known.

In this study the initial copeptin median was 6.3 pmol/L. To date, no internationally accepted normal copeptin values exist, but published values for healthy individuals are in the range 2–10 pmol/l [14, 28–30]. Our result could be seen as to be within the normal range and thus not being elevated, but because we observed a gradual decrease over time (-26%), it is suggestive that the median copeptin values found in our study were at least slightly elevated compared to the assumed baseline values in healthy children. The dynamics were not as anticipated, as we have seen in a cytokine analysis of the ProPAED cohort that pro- and anti-inflammatory cytokines decreased very rapidly over the study period [24], so a steeper copeptin decrease was expected. Especially, if the regulatory swiftness is taken into account, with which vasopressin and copeptin plasma values change when regulating body water homeostasis: copeptin decreases with a half-life of only 26 minutes after oral water challenge [14, 28, 31,

32]. Copeptin is actively released via the activation of the endocrine stress axis at equimolar concentrations from the posterior pituitary into circulation in situations with an increased plasma osmolality, hypovolemia and high individual stress level. The stress axis activation results in release of vasopressin, adrenocorticotropic hormone and cortisol [29, 32, 33]. In this context dehydration as a potential source of bias has to be taken into account. Hydration assessment of the included patients was not mandatory and as no blood gas analyses or metabolic assessments were recorded, a lack of adequate fluid intake cannot be excluded as possible confounder. However, an inflammation driven persistent and gradually decreasing stress level was more likely in our cohort than dehydration and we assume that the copeptin course in our study rather reflected the stress reaction of the body to the inflammatory state than body water homeostasis disturbance.

Besides kinetics, associations with potential markers of disease severity were of interest for this study but for copeptin we found none. Several studies have shown copeptin to be related to severity and complications of LRTI. Du et al. investigated plasma copeptin levels in 265 children and found median levels of 73.0 pmol/L in complicated pneumonia in 2013 [15]. This association of copeptin with complicated LRTI was later confirmed by two further studies [16, 19]. In our case we assume that the number of patients with LRTI complications (n = 5, S3 Table) was just too low to create significant influence in the model. Therefore, we cannot draw a rational conclusion on copeptin as marker of disease severity. Gender has been described as potential factor influencing copeptin levels [14, 34], but in our cohort, we did not find any gender disparity. Most likely, because the inflammatory state overruled a possible gender disparity in unstressed individuals.

The initial median MR-proADM concentrations in the present cohort were higher than the median values for healthy children reported by Michels et al. (0.22 nmol/L) and Hauser et al. (0.34 nmol/L) [35, 36], suggesting a possible role for identifying individuals with more severe LRTI and thus need of antibiotics. In contrast to copeptin, adrenomedullin is actively involved as a hormokine in inflammatory reactions and triggers cytokine release [37, 38], but is also triggered by cytokines themselves, most importantly by IL-1β and TNF [39]. However, the MR-proADM values stayed elevated longer than the cytokines measured in the ProPAED cohort and did consecutively not correlate with them [24]. When analyzing the relationship between cytokines and markers as copeptin and MR-proADM in the present cohort of pediatric LRTI, one has to keep in mind, that the children and adolescents attended the emergency room with a history of fever lasting (median) three days. Therefore, it is most likely, that the initial up-stream inflammatory activation happened, at least in part, before the first measurements of cytokines and inflammatory biomarkers were performed for the ProPAED study.

In recent studies MR-proADM was able to predict LRTI complications, such as pleural effusions and bacteremia [18, 20, 21]. In the present study IV administration of antibiotics was associated with higher MR-proADM concentrations on day 1, and ICU admission was associated with a slow increase in MR-proADM, indicating a persistent inflammatory state. Although IV administration of antibiotics and ICU admission can very well be triggered by other factors than severe invasive bacterial disease, these two markers can be seen as surrogate markers for disease severity. Our findings are in line with previous studies. Alcoba et al. found MR-proADM to be associated with bacteremia and empyema [18], and Sardà Sánchez et al. found MR-proADM related to pneumonia with pleural effusion (no information on empyema provided), but not to ICU admission, as in our case. However, in that study, only hospitalized patients were analyzed and in our study we included in- and outpatients, therefore our cohort might have been less sick than the cohort of Sardà Sánchez et al. [18, 21]. Further, the most recent study by Florin et al. [20], evaluating also in- and outpatients, was in line with our and former results: the association of MR-proADM with higher severity of disease was significant.

In that study ICU admission was also significantly associated with higher MR-proADM values. Last, blood culture positivity was associated with a more distinct MR-proADM decrease in our study. This is most likely due to the effective antibacterial treatment of streptococci, the only pathogen found in blood cultures in our study, being highly susceptible to empirical treatment.

## Limitations

The present study was performed in Switzerland, a highly developed country with an excellent medical safety netting, in two emergency departments of pediatric tertiary centers where generally the rate of complicated medical illness is low. The rate of complications from LRTI here was low and no strong correlation of complications to copeptin or MR-proADM concentrations could be established. In a different setting (e.g. different medical system or inclusion only of hospitalized children) with a higher prevalence of complications the result might be different and a stronger impact of LRTI complications on copeptin and MR-proADM kinetics could result. Another limitation is the lack of a healthy pediatric control group that would have been necessary to establish comparable pediatric normal values. In our case, we could only refer to previously published values in adults and children. Chest radiographs were not mandatory in the ProPAED study and only the more severe cases underwent chest radiographs. To avoid selection bias we did not perform any diagnostic accuracy analysis for the prediction of chest x-ray confirmed pneumonia or other types of radiographically diagnosed LRTI.

In summary, this study evaluated kinetics of copeptin and MR-proADM and potential factors influencing the course over time. Our study supports the potential value of MR-proADM plasma concentrations as a prognostic factor for severe pediatric LRTI. Plasma copeptin concentration had no added value in our setting. These results call for larger studies confirming the clinical use of MR-proADM in pediatric LRTI and studies on copeptin in settings with more serious febrile LRTI.

## Supporting information

**S1 Appendix.**
(DOCX)

**S1 Fig. Age group and biomarkers.** Age-stratification and change in MR-proADM (nmol/L), and copeptin (pmol/L) concentrations over the study days 1, 3, and 5.
(DOCX)

**S1 Table. Cut-off values to define normal or elevated breath and heart rates according to patient's age.** Classification as normal or elevated rate were set according Fleming S, Thompson M, Stevens R et al. Normal ranges of heart rate and respiratory rate in children from birth to 18 years of age: a systematic review of observational studies. The Lancet. 2011; 377 (9770):1011±8 and WHO.
(DOCX)

**S2 Table. Age group and biomarkers.** Age-stratification and change in MR-proADM (nmol/ L), and copeptin (pmol/L) concentrations over the study days 1, 3, and 5. SD: Standard deviation.
(DOCX)

**S3 Table. Type and day of complication experienced by five patients during the study period of 5 days.** ICU: intensive care unit; ARDS: acute respiratory distress syndrome. (DOCX)

**S4 Table. Copeptin, and MR-proADM, concentration profiles and relative change between study days.** MR-proADM: pro-adrenomedullin; IQR: inter-quartile range. (DOCX)

## Acknowledgments

ProPAED study group membership: Arnold Amacher, Gerald Berthet, Heiner C. Bucher, Sabine Faisst, Muriel Gambon, Juerg Hammer, Beat Mueller, Diana Reppucci, Juliane Schaefer, Jacques Schneider, Philipp Schuetz, Jody Staehelin-Massik, Daniel Trachsel, Urs B. Schaad, Edda Weimann, Urs Zumsteg. Lead author of this group is Dr. Jan Bonhoeffer (jan.bonhoeffer@ukbb.ch).

## Author Contributions

**Conceptualization:** Aline Fuchs, Ulrich Heininger, Gabor Szinnai, Jan Bonhoeffer.

**Data curation:** Aline Fuchs.

**Formal analysis:** Philipp Baumann, Aline Fuchs, Verena Gotta, Gabor Szinnai, Jan Bonhoeffer.

**Funding acquisition:** Ulrich Heininger, Jan Bonhoeffer.

**Investigation:** Philipp Baumann, Nicole Ritz, Gurli Baer, Jessica M. Bonhoeffer, Michael Buettcher, Ulrich Heininger, Gabor Szinnai, Jan Bonhoeffer.

**Methodology:** Ulrich Heininger, Gabor Szinnai, Jan Bonhoeffer.

**Project administration:** Ulrich Heininger, Jan Bonhoeffer.

**Resources:** Ulrich Heininger, Jan Bonhoeffer.

**Software:** Aline Fuchs, Jan Bonhoeffer.

**Supervision:** Gurli Baer, Ulrich Heininger, Jan Bonhoeffer.

**Validation:** Philipp Baumann.

**Visualization:** Aline Fuchs.

**Writing – original draft:** Philipp Baumann.

**Writing – review & editing:** Philipp Baumann, Aline Fuchs, Verena Gotta, Nicole Ritz, Gurli Baer, Jessica M. Bonhoeffer, Michael Buettcher, Ulrich Heininger, Gabor Szinnai, Jan Bonhoeffer.

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
