## [Decision Letter · Decision Letter 0]

11 Nov 2020

PONE-D-20-08646

Copeptin and mid regional proadrenomedullin (MR-proADM) in pediatric lower respiratory tract infections

PLOS ONE

Dear Dr. Baumann,

Thank you for submitting your manuscript to PLOS ONE. After careful consideration, we feel that it has merit but does not fully meet PLOS ONE’s publication criteria as it currently stands. Therefore, we invite you to submit a revised version of the manuscript that addresses the points raised during the review process.

We look forward to receiving your revised manuscript.

Kind regards,

Julia Robinson

Senior Editor

PLOS ONE

Journal Requirements:

2. One of the noted authors is a group or consortium [ProPAED study group]. In addition to naming the author group and listing the individual authors and affiliations within this group in the acknowledgments section of your manuscript, please also indicate clearly a lead author for this group along with a contact email address.

Reviewers' comments:

Reviewer's Responses to Questions

**Comments to the Author**

1. Is the manuscript technically sound, and do the data support the conclusions?

Reviewer #1: Partly

Reviewer #2: Yes

Reviewer #3: Partly

2. Has the statistical analysis been performed appropriately and rigorously? 

Reviewer #1: Yes

Reviewer #2: Yes

Reviewer #3: Yes

3. Have the authors made all data underlying the findings in their manuscript fully available?

Reviewer #1: Yes

Reviewer #2: Yes

Reviewer #3: Yes

4. Is the manuscript presented in an intelligible fashion and written in standard English?

Reviewer #1: Yes

Reviewer #2: Yes

Reviewer #3: Yes

5. Review Comments to the Author

Reviewer #1: This is a well conducted and reported study. Nevertheless I have some comments and suggestions about how the manuscript could be further improved.

In the abstract, statements about differences in concentrations of MR-proADM in lines 43 to 45 should include the estimated difference and 95% confidence interval, not just a p-value. Additionally, the statements about association of copeptin with clinical characteristics or complications should be unambiguous, and terms such as "relevant association" and "correlated moderately" should be avoided.

The description of the non-linear mixed model in the methods has been done well. The descriptions in line 172 to 179 is unclear; 'parameter P' is mentioned without any introduction as to what it represents, and it is unclear what 'COV' means. Additionally there is a typo on line 172 ('median' not 'mdeian'?). Additionally, it is unclear what kind of models were used to address the classification question of predicting antibiotic treatment (lines 180 to 187). More details should be included here.

Most importantly, I think the study would benefit from a better approach to selecting the covariates to be included in the models for the various outcomes. This selection should consider whether the covariates would plausibly be related to the outcomes. For example it doesn't seem plausible to explore association between administration of intravenous antibiotics and baseline levels of the biomarkers, because the baseline implies before treatment i.e. pre-antibiotics, pre-blood culture, etc.

Lastly, the reporting of the results should focus on explaining what the various parameters of the non-linear model mean, rather than just citing their numerical values.

Reviewer #2: The authors of this manuscript reported the possible roles of two novel biomarkers (copeptin and MR-proADM) in predicting disease severity in pediatric lower respiratory infections (LRTIs). It was a post-hoc analysis of a randomized controlled trial evaluating procalcitonin (PCT) as a biomarker guiding antibiotic treatment in children and adolescents with LRTIs in the ProPAED trial. Biomarker research aimed at improving disease diagnosis and prognostication remains a relevant issue in scientific literature. Although the authors gave an in-depth report of their study which made a case for MR-proADM alone or in combination with PCT, cytokines or C-reactive protein (CRP) in predicting disease severity in LRTIs, there are few specific concerns that I request them to address to enhance the manuscript.

Reviewer #3: This study is a subanalysis of a prior RCT of children with LRTI that focuses on biomarker kinetics in pediatric LRTI. The strength of this manuscript lies in its novelty – the kinetics of most of these biomarkers in pediatric CAP has not been reported previously. There are several challenges to the approach and presentation of data in this manuscript that make the conclusions difficult to understand.

MAJOR ISSUES

1. This is a heterogeneous population – children with bacterial pneumonia require antibiotics and children with bronchiolitis do not. Therefore, to include use of IV antibiotics as one of the primary outcomes is not appropriate for half of the already small study population. This study requires stratifying data into those with and without pneumonia

2. The story is somewhat confusing as written – is this about severe outcomes? Use of antibiotics? Only copeptin and proADM or really about 4 biomarkers? I would recommend that this focus on kinetics of CRP, PCT, copeptin and proADM. The title suggests that it is only copeptin and proADM but there is important data presented on other biomarkers as well that should be accounted for in the title and results sections.

3. Results and tables require clarity on the meaning of the statistical parameters. The message is lost because of a focus on statistical rather than clinical terminology throughout.

ABSTRACT

1. I’d recommend that the results section include more numerical results. It would be more meaningful for readers to see effect estimates with confidence intervals rather than simply p-values.

2. If the conclusion is that there is additional clinical value using combinations of biomarkers, I would recommend giving some numerical results of the correlations in the results.

INTRODUCTION:

1. Would be more specific in the citation placement of references 1-6. The individual references should be placed after the prior sentences, as they all do not support the sentence that up to 90% are unnecessarily prescribed antibiotics. Would consider adding the following reference to support this statement: Florin TA et al. JPIDS 2020;9(2).

2. An important reference that should be included in Lines 55-57 is: Stockmann C et al. JPIDS 2018;7(1).

3. A more recent reference regarding MR-proADM and disease severity that should be included in the introduction is: Florin TA et al. Clin Infect Dis 2020 Aug 6; ciaa1138

METHODS

1. This data is now more than 10 years old and occurred during the H1N1 influenza pandemic. Can the authors comment on the age of the data and the possible influence of the pandemic on the results?

2. Line 109 – was pneumococcus the only bacterium considered for systemic bacterial infection? What about other known bacterial pathogens of pneumonia?

3. Line 114 – it is not clear what age-dependent reference standard is used for vital signs with the exception of respiratory rate. This is not laid out in the text or in Table S1.

4. Line 116 – the Fleming percentiles were derived from healthy children in the outpatient setting, thus I’m not certain they are appropriate in this sick hospitalized population. Others (Bonafide, Daymont, Oostenbrink, Thompson) have published vital sign curves for ED and hospitalized children which would be more appropriate to use.

5. Lines 145-179 – the section on modeling is not intuitive or easily understood by the average medical reader. I recommend that this entire section be written to summarize the statistical methods in prose that allows the average clinician reading this paper to understand what was performed in the body of the main manuscript. The details of the modeling can be expounded on in S1 Appendix.

6. Lines 166-169 – can the authors justify their use of a stepwise approach to covariate selection, rather than an approach based on biological and clinical plausibility?

7. Lines 182-183 – it appears that discriminating IV antibiotics vs oral or no antibiotics is a primary outcome. Were there standardized criteria for who received IV antibiotics? This is critical, as this otherwise becomes reflective more of an individual clinician’s gestalt or the fact that an IV line was present rather than a true need for IV antibiotics. This is what makes use of IV antibiotics a challenging primary outcome.

RESULTS

1. Line 207 – inclusion criteria makes it sound as if only those who had all measurements on day 1, 3, and 5 were included. If this is the case, I am concerned about selection bias for this population, as most children with LRTI who are hospitalized stay in the hospital for fewer than 5 days (median ~2-2.5 days per literature).

2. Line 215 – there are important data that are buried in Table S1. This table is labeled as a list of variables or data dictionary, when in fact the last column is important data to describe the study population and outcomes. This data should be in the main body of the paper.

3. Line 224 - Less than half the children in this study had pneumonia. The pathophysiology and microbiology of different lower respiratory tract infections are different, and therefore I’m concerned that this is a highly heterogeneous population (with a relatively small n, in addition). Children with bronchiolitis shouldn’t be requiring antibiotics to begin with and therefore it is not appropriate to include them with children with pneumonia in a primary outcome that includes IV antibiotic use.

4. Lines 246-252 – this section needs to be edited to explain the results in prose rather than relying on statistical parameters. The meaning of these results is lost in statistical jargon.

5. Lines 264-266 – same as above – rather than talking about ‘baseline parameter’ and ‘slope parameter,’ this would be easier to read and interpret if the text read: “the following variables were statistically associated with initial levels of MR-proADM…ICU admission and microbiology results were associated with the change in MR-proADM from baseline to Day 5.” (if I’m interpreting the statistical jargon appropriately)

6. Table 2 and 3 – it would be helpful to have footnotes explaining the parameters in this table. What is meant by ‘limit?’

7. Line 302 – predict is the wrong term here. The biomarker is not doing the predicting – I would stick with MR-proADM discriminated those children receiving IV antibiotics vs those who did not. If less than half of the children in this study had pneumonia and hospitalized children with pneumonia generally warrant antibiotics, aren’t these results really a proxy for discriminating pneumonia from no pneumonia? There should be an examination of the interaction of diagnosis and IV antibiotics in these analyses.

8. Figure 2 – requires labels

DISCUSSION

1. Line 333 – the discussion states that decreases were not as steep for CRP and PCT, yet the declines observed in Fig S1 do suggest that there were in fact declines in CRP and PCT. If this statement is going to be made, I would recommend some presentation of data in the Results section. In fact, I think that this manuscript would benefit from being reframed as biomarker kinetics in general, including CRP, PCT, copeptin and proADM, as that is really what this manuscript is about – not just copeptin and proADM.

2. Line 385 – include some discussion of Florin TA et al. Clin Infect Dis 2020 Aug 6; ciaa1138

3. Line 400-401 – just because the correlations were moderate here (some were not very good at all), one cannot conclude that combinations of biomarkers may be helpful. It would have been helpful if there was statistical analysis that focused on the outcomes of interest with combinations of biomarkers to be able to make this claim rather than simple correlations.

6. PLOS authors have the option to publish the peer review history of their article (what does this mean?). If published, this will include your full peer review and any attached files.

Reviewer #1: No

Reviewer #2: **Yes: **Samuel Uwaezuoke

Reviewer #3: No

---

## [Author Response · Author response to Decision Letter 0]

2 Apr 2021

Dear Julia Robinson,

thank you for considering the publication of our manuscript and the thorough review. Here we answer point-by-point to all aspects raised by the reviewers. Please see the corresponding corrections in the uploaded manuscript which we hope is now suitable for publication. 

All line numbers in the revised manuscript correspond to the track change version.

Journal Requirements:

We carefully checked our manuscript again. All requirements that are provided in these two pdf-files are met. 

2. One of the noted authors is a group or consortium [ProPAED study group]. In addition to naming the author group and listing the individual authors and affiliations within this group in the acknowledgments section of your manuscript, please also indicate clearly a lead author for this group along with a contact email address.

The lead author for this group is Dr. Jan Bonhoeffer, we added his contact details to the Acknowledgments section (lines 545/546oo).

Summary

Before we begin with the point-for-point answers for each reviewer we would like to introduce a short section-for-section summary, as some reviewers requested important major changes that affected the whole direction of the article and are likewise important to all three reviewers. We would like to summarize these changes here and come back to them in detail later.

Abstract and Introduction:

We thank the reviewers for the advice to focus on the clinical relevance of our work and to favor language that is more intuitively understandable also without biostatistical degree. Therefore the abstract was reworded with respect to reviewers` requirements. We trust that the abstract is now appropriate for clinical physicians. We avoided ambiguous terms and present more data in numerical detail.

The introduction was shortened, partly reworded and restructured to emphasize the focus on the clinical aspects and the primary outcome biomarkers` kinetics. 

Methods/Results:

Parts of the methods/results sections were outsourced to S1 Appendix, language was adapted to become easier to read. ROC analysis for iv administration of antibiotics was deleted due to the rather vague outcome parameter “administration route”.

Discussion: 

The discussion was in large parts reorganized and re-structured according to the requirements of reviewer #2. It is now also significantly shorter.

Conclusion: 

The conclusion has been changed according to the modifications made to the article.

Funding: We added a funding part as test kits were provided by BRAHMS free of charge.

Reviewer #1: This is a well conducted and reported study. Nevertheless I have some comments and suggestions about how the manuscript could be further improved.

We are grateful to reviewer # 1 for his comments relevant to the improvement of this manuscript. According to his suggestion, we have changed the following sections.

In the abstract, statements about differences in concentrations of MR-proADM in lines 43 to 45 should include the estimated difference and 95% confidence interval, not just a p-value. Additionally, the statements about association of copeptin with clinical characteristics or complications should be unambiguous, and terms such as "relevant association" and "correlated moderately" should be avoided.

Abstract: The estimated differences and the change per day of the biomarker values have been entered into lines 47-52. Potentially ambiguous statements have been deleted (line 52 - 54).

Additionally there is a typo on line 172 ('median' not 'mdeian'?). 

Methods: Typo was corrected, thank you.

Additionally, it is unclear what kind of models were used to address the classification question of predicting antibiotic treatment (lines 180 to 187). More details should be included here.

Methods: For the sake of clarity we deleted the ROC analysis (Fig 2) and the corresponding S6 Table. IV administration is a rather weak outcome parameter as it could be influenced by other factors (iv line in place, need of fluids, etc.) and as it was correctly stated by the reviewers.

Most importantly, I think the study would benefit from a better approach to selecting the covariates to be included in the models for the various outcomes. This selection should consider whether the covariates would plausibly be related to the outcomes. For example it doesn't seem plausible to explore association between administration of intravenous antibiotics and baseline levels of the biomarkers, because the baseline implies before treatment i.e. pre-antibiotics, pre-blood culture, etc.

Methods: For the selection of covariates we had different demographic, clinical and laboratory parameters at hand (age; sex; body temperature; breath rate; heart rate; days of fever before inclusion; complications including sepsis, pleural effusion, pleural empyema, ARDS; hospital or ICU admission; oxygen requirement; form of antibiotic administration and antibiotic pre-treatment; CRP; PCT; nasopharyngeal aspirates; blood culture; and diagnosis given by the attending physician), see Table 1. Our exploratory approach was to include these into the base model to evaluate broadly the impact on MR-proADM and copeptin kinetics. IV-administration stood out (among others) even though a part of the population was pre-treated (15%). A potential biomarker suggesting need or withhold of antibiotic treatment should be discriminative, even though some patients would have been pre-treated already. Therefore we retained this parameter in the final model. 

We agree, that this approach is rather broad, but as published data for respective associations with copeptin and MR-proADM kinetics in children is scarce, we still believe, that this is appropriate.

Lastly, the reporting of the results should focus on explaining what the various parameters of the non-linear model mean, rather than just citing their numerical values

Results: In the results section, we would like to state just the numerical results and comment on these in the discussion. However, this also refers to other review comments criticizing the language we used. We changed this throughout the article and feel that also the results section is now easier to understand. 

Reviewer #2: General comments

The authors of this manuscript reported the possible roles of two novel biomarkers (copeptin and MR-proADM) in predicting disease severity in pediatric lower respiratory infections (LRTIs). It was a post-hoc analysis of a randomized controlled trial evaluating procalcitonin (PCT) as a biomarker guiding antibiotic treatment in children and adolescents with LRTIs in the ProPAED trial. Biomarker research aimed at improving disease diagnosis and prognostication remains a relevant issue in scientific literature. Although the authors gave an in-depth report of their study which made a case for MR-proADM alone or in combination with PCT, cytokines or C-reactive protein (CRP) in predicting disease severity in LRTIs, there are few specific concerns that I request them to address to enhance the manuscript.

We are grateful to the reviewer #2 for his help to improve this article.

Specific comments

1. Abstract and Title: The full title of your manuscript does not convey the main objective of the study. Curiously, the title is the same as your short title. Based on the aim of your study, I suggest a modification of your full title to read- ‘The kinetic profiles of copeptin and mid-regional proadrenomedullin in pediatric lower respiratory tract infections.’ 

Title: We changed the title according to his suggestion.

In the background of your abstract, you mentioned that both novel biomarkers may be predictive of complications of pediatric LRTIs. Which complications are you referring to? 

Abstract: In fact, we did not explain LRTI-complications in the abstract. We defined complications in the variables section of the methods (parapneumonic effusion, empyema, ARDS, sepsis, shock) in lines 119-120. While rewording with the focus on biomarkers` kinetics and the impact of clinical and laboratory features, we choose to temper the part of predicting complications and omitted this sentence in the final version of the abstract.

Again, the phrase-‘The change over time of these biomarkers during LRTIs’ may better be rendered thus- ‘The kinetics of these biomarkers during LRTIs.’ 

Abstract: We changed this according to this suggestion (line 31).

The second part of the aim read- ‘…and to investigate the relationship between copeptin, MR-proADM and demographic, clinical and laboratory characteristics of pediatric LRTI.’ I suggest you rephrase this for clarity. 

Abstract: We changed this sentence according to his and the overall suggestion to focus more on clinical relevant aspects, it now reads: “We aimed to analyze kinetic profiles of copeptin and MR-proADM and the influence of clinical and laboratory aspects on biomarkers` progression.” (lines 34-37).

2. Introduction: Study aims stated in lines 79-84 appear slightly different from the aims under Abstract. Can you please harmonize them?

We changed the phrases accordingly.

Abstract: “We aimed to analyze kinetic profiles of copeptin and MR-proADM and the impact of clinical and laboratory factors on biomarkers` progression.”

Introduction: “Therefore, we aimed at (i) determining the kinetic profiles of copeptin and MR-proADM over five days in children and adolescents admitted to the emergency department for suspected LRTI, and (ii) investigating the influence of clinical and laboratory covariates as well as the development of LRTI complications on copeptin and MR-proADM course over time in a retrospective subanalysis of the ProPAED trial.”

3. Methods: Every study methodology should have clarity and easy reproducibility as key features. Why did you adopt a post-hoc analysis of an RCT in your study? Post-hoc analysis conducted and interpreted without sufficient consideration of the ‘multiple testing problem’ is criticized by some scholars who believe the statistical associations are often spurious. In fact, you to admitted this methodological flaw of using a retrospective approach in one of your listed study limitations (lines 428-430)

Methods: We totally agree on this comment. We acknowledge this possible flaw, but can state, that our main aim was to demonstrate the kinetics of Pro-ADM and copeptin over 5 days in a well-defined population. Comparable data was missing so far. Besides the plain demonstration of the kinetics we wanted to go beyond and try to finds answers to the question of possible causality. Obviously this approach can only produce loose associations and no clear causal reasons for biomarkers` kinetics.

To clarify this approach, we inserted a statement on this in the beginning of the discussion, lines 367-369:

“Our main aim was to determine the dynamics of these biomarkers over the study period of five days. Our additional aim was to test for impact of clinical and laboratory markers that might indicate severity of disease on the biomarkers kinetics.”

4. Discussion: There are many redundant statements in your discussion which made it difficult to follow and understand. For instance, there are repetition of results in several paragraphs. I suggest a discussion with this frame work for clarity and conciseness: 

Discussion: We restructured the discussion according to the frame work the reviewer suggested it below. We added a small introductory paragraph for those readers who directly start reading in the discussion. Throughout, we deleted any information that was already present in the other paragraphs of the article (e.g. lines 374-382) except for direct comparisons of our values with published values.

paragraph 1: the research problem or gap and the significance of addressing or filling it 

Discussion: We formulated the need for this study and inserted this paragraph into lines 383-389.

paragraph 2: Critical analysis of your major findings +

paragraph 3 additional findings and how they fit into the existing literature (paragraph 3), 

Discussion: 

We choose to combine these paragraphs, so that we could discuss first copeptin (major + additional findings) and then MR-proADM (major + additional findings). As such, the requested structure could be achieved, just for the two biomarkers one following the other. The whole paragraph was inserted for copeptin in lines 390-443 and for MR-proADM in lines 445-478.

study limitations (paragraph 4), 

Discussion: Was presented before already as an own paragraph, is now slightly modified in the lines 510-526.

future directions (paragraph 5) 

This was briefly integrated into the summary (conclusion, lines 533-534) at the end of discussion. 

and overall conclusion and major impact of the study (paragraph 6). 

Discussion: Conclusion has been adjusted according to the changes made throughout the article (now lines 528-534.

Your study limitations may appear illogical to some readers. In lines 423-425, you tend to suggest that severity of the presentation of LRTIs is determined by the location of medical practice. However, it is obvious there are underlying conditions such as immunosuppression that could worsen disease presentation and serve as confounders. In any case, your study exclusion criteria lent credence to this fact (lines 94-95).

Limitations: This is true, underlying conditions like immunosuppression could definitively influence the disease progress and severity. We excluded these cases and did not have any patients with known immune insufficiency in the cohort. However, in our region of Europe it is obvious, that parents can seek medical advice within minutes and around the clock. This is definitively not the case in other parts of the world. Therefore, we are careful with reproducibility of our study in other medical systems.

Reviewer #3: This study is a subanalysis of a prior RCT of children with LRTI that focuses on biomarker kinetics in pediatric LRTI. The strength of this manuscript lies in its novelty – the kinetics of most of these biomarkers in pediatric CAP has not been reported previously. There are several challenges to the approach and presentation of data in this manuscript that make the conclusions difficult to understand.

We thank also reviewer #3 for the extensive review of our article and the amount of time the reviewer invested into it. We have redirected the general orientation of the article to meet the requested requirements.

MAJOR ISSUES

1. This is a heterogeneous population – children with bacterial pneumonia require antibiotics and children with bronchiolitis do not. Therefore, to include use of IV antibiotics as one of the primary outcomes is not appropriate for half of the already small study population. This study requires stratifying data into those with and without pneumonia

In general: Our primary objective was to display biomarkers` kinetics and not biomarker guided antibiotic therapy. We are now aware that this was not as clear as we intended it to be. We did a thorough workover to emphasize kinetics and temper the statements about the associations with clinical and other lab values. We wanted to display biomarkers` kinetics for the whole cohort of pediatric patients with LRTI as we think, that the differentiation of LRTI into subgroups (pneumonia, bronchiolitis, bronchitis) has several flaws and diagnoses can relevantly overlap or dynamically develop from one into another. However, for the sake of completeness, in the article version submitted firstly, we actually did stratify the patients into diagnosis groups (presented in the text and formerly S1 Table, now integrated in Table 1 in main article body) but the stratification did not reveal any relevant associations and was not retained in the final model. Further, only febrile children were included into the initial study. Fever is still a strong driver for antibiotic prescription because of concerns for primary/secondary infection and therefore we wanted to include all children into the final model.

This said, the reviewer is correct, that IV antibiotics should not be a primary outcome and we did not intended it to be primary outcome. It was an association that we found when examining factors that could potentially impact biomarkers` course over time and we think, it is important to mention it. However, as just being an association and as IV administration can indeed be driven by factors other than severe illness, we agree to not overemphasize this.

2. The story is somewhat confusing as written – is this about severe outcomes? Use of antibiotics? Only copeptin and proADM or really about 4 biomarkers? I would recommend that this focus on kinetics of CRP, PCT, copeptin and proADM. The title suggests that it is only copeptin and proADM but there is important data presented on other biomarkers as well that should be accounted for in the title and results sections.

In general: This article is about copeptin and pro-ADM kinetics and their association with clinical and laboratory covariates. In fact, we initially added PCT and CRP kinetics for comparative reasons, but did not explore them the same way as we did with copeptin and pro-ADM. This indeed was confusing. We deleted these graphs and focused the whole article entirely on copeptin and pro-ADM. PCT and CRP will be evaluated along with other factors in a separate article, those factors would be out of scope of this article.

3. Results and tables require clarity on the meaning of the statistical parameters. The message is lost because of a focus on statistical rather than clinical terminology throughout.

Results: This was mentioned by reviewer #1 as well and we are grateful for these comments. We changed the jargon throughout the article and tables to make it more accessible for the broad readership.

ABSTRACT

1. I’d recommend that the results section include more numerical results. It would be more meaningful for readers to see effect estimates with confidence intervals rather than simply p-values.

Abstract: This corresponds well to the comment of reviewer #1, who also requested more numerical values. We have entered the estimated differences between biomarker and decrease values as numerical data into lines 47-52.

2. If the conclusion is that there is additional clinical value using combinations of biomarkers, I would recommend giving some numerical results of the correlations in the results.

Abstract: We agree, that correlation data were missing, but as correlations were rather weak and as we stepped back from the statement on usefulness of combined biomarkers, we deleted the two respective sentences.

INTRODUCTION:

1. Would be more specific in the citation placement of references 1-6. The individual references should be placed after the prior sentences, as they all do not support the sentence that up to 90% are unnecessarily prescribed antibiotics. Would consider adding the following reference to support this statement: Florin TA et al. JPIDS 2020;9(2).

Introduction: References have been changed accordingly and the cited reference was inserted into line 65.

2. An important reference that should be included in Lines 55-57 is: Stockmann C et al. JPIDS 2018;7(1).

Introduction: This reference was inserted into line 66.

3. A more recent reference regarding MR-proADM and disease severity that should be included in the introduction is: Florin TA et al. Clin Infect Dis 2020 Aug 6; ciaa1138

Introduction: Indeed, this article was not published, when we submitted our article. We are grateful to pick this up in line 79 and in the discussion in line 471.

METHODS

1. This data is now more than 10 years old and occurred during the H1N1 influenza pandemic. Can the authors comment on the age of the data and the possible influence of the pandemic on the results?

First, the analyses were performed in 2016 (6 years storage at -80°C) and as there is no hint for long-term degredation at this temperature (Copeptin: Heida JE, Boesten LSM, Ettema EM, Muller Kobold AC, Franssen CFM, Gansevoort RT, et al. Comparison of ex vivo stability of copeptin and vasopressin. Clin Chem Lab Med. 2017;55(7):984-992; MR-proADM: Morgenthaler NG, Struck J, Alonso C, Bergmann A. Measurement of Midregional Proadrenomedullin in Plasma with an Immunoluminometric Assay. Clin Chem. 2005;51(10):1823-1829) we think, that the samples were still usable. Second, H1N1 is assumed to drive inflammatory biomarkers as usual in influenza viruses: they are associated with severe outcome (Vasileva D, Badawi A. C-reactive protein as a biomarker of severe H1N1 influenza. Inflamm Res. 2019;68(1):39-46). This was shown in particular for MR-proADM in inflenza patients (Valero Cifuentes S, García Villalba E, Alcaraz García A, Alcaraz García MJ, Muñoz Pérez Á, Piñera Salmerón P, et al. Prognostic value of pro-adrenomedullin and NT-proBNP in patients referred from the emergency department with influenza syndrome. Emergencias : revista de la Sociedad Espanola de Medicina de Emergencias. 2019;31(3):180-184).

Thus, we think, that a patient enrolled during the pandemic H1N1-influenza situation at that time should be examined for biomarkers the same way as any other influenza patient in different years. Severe disease is expected to drive inflammatory biomarkers.

2. Line 109 – was pneumococcus the only bacterium considered for systemic bacterial infection? What about other known bacterial pathogens of pneumonia?

Methods: No, pneumococcus was not the only bacterium considered for systemic bacterial infection. In lines 108 – 110 of the original version, we mentioned, that we classify microbiology findings into 4 categories: “Microbiology results were classified in 4 categories: not performed, negative, systemic bacterial infection (blood culture positive for pneumococcus/streptococcus = invasive pneumococcal infection) and other.” Indeed, we had overall 6 patients with positive blood culture, in one patient growth of coagulase negative Staphylococcus was present, this was deemed contamination (no intravascular access or foreign materials present) and the patient was treated as blood culture negative. We did consider, of course, many bacterial pathogens that could have grown from blood culture, but we just found the above mentioned ones.

3. Line 114 – it is not clear what age-dependent reference standard is used for vital signs with the exception of respiratory rate. This is not laid out in the text or in Table S1.

S1 Table: This is correct. Reference values were laid out in S2 Table, however the link was indeed missing in S1 Table. S1 Table was merged with Table 1 as it was required to be part of the main text body in one of the next comments, there we refer to S1 Table in line 238. 

4. Line 116 – the Fleming percentiles were derived from healthy children in the outpatient setting, thus I’m not certain they are appropriate in this sick hospitalized population. Others (Bonafide, Daymont, Oostenbrink, Thompson) have published vital sign curves for ED and hospitalized children which would be more appropriate to use.

Methods: Thank you for the comment and the recommendations for alternative percentiles. We would like to stick to the Fleming percentile because of the following reasons: 

• The Fleming cohort is large (Fleming: 150 080 HR and 7565 RR measurements in healthy children, Bonafide (2013): 77 825 HR and 77 825 RR measurements in children hospitalized in tertiary care, Daymont (2015): 60 863 observations (HR only) in children hospitalized in tertiary care, Oostenbrink (2012): 1555 observations in ED (RR only), Thompson (2008): 1933 observations in primary care (HR only). 

• Most important, we wanted healthy children as reference to which we could compare vital sign values from the study.

• In our cohort 51% of patients were hospitalized at Day 1 and 49% returned home and were seen in the outpatient clinics on Day 3 and Day 5. Many (21%) of the patients hospitalized at Day 1 returned home before Day 3 or Day 5. Thus, more vital sign measurements were recorded in the ambulatory setting and not in the hospitalized setting.

5. Lines 145-179 – the section on modeling is not intuitive or easily understood by the average medical reader. I recommend that this entire section be written to summarize the statistical methods in prose that allows the average clinician reading this paper to understand what was performed in the body of the main manuscript. The details of the modeling can be expounded on in S1 Appendix.

Methods: We agree on this. In fact, the information on base model was already redundantly in place in S1 Appendix, so it was deleted from the article. Further, the explanations on covariate modelling have been moved to S1 Appendix as well, leaving a plain language summary in the article text. 

6. Lines 166-169 – can the authors justify their use of a stepwise approach to covariate selection, rather than an approach based on biological and clinical plausibility?

Methods: A preliminary selection of covariates based on biological and clinical plausibility was first performed. The selected covariates were then assessed for statistical significance using a standard stepwise approach.

7. Lines 182-183 – it appears that discriminating IV antibiotics vs oral or no antibiotics is a primary outcome. Were there standardized criteria for who received IV antibiotics? This is critical, as this otherwise becomes reflective more of an individual clinician’s gestalt or the fact that an IV line was present rather than a true need for IV antibiotics. This is what makes use of IV antibiotics a challenging primary outcome.

Methods: Biomarkers` kinetics and not IV antibiotic use was the primary outcome. We reworded the article to clearly state this. IV administration was performed according to attending physicians´ discretion. We are aware, that IV use is not a very precise marker for disease severity because of several points that can influence the decision to give antibiotics intravenously (IV line in place, in need of IV fluids etc.). Therefore, we do not want to overemphasize this. However, it stays in the pool of possible associations to test for, because children that get antibiotics intravenously can generally be considered to be sicker than those without IV administration. Further, we deleted the ROC analysis for the same reason.

RESULTS

1. Line 207 – inclusion criteria makes it sound as if only those who had all measurements on day 1, 3, and 5 were included. If this is the case, I am concerned about selection bias for this population, as most children with LRTI who are hospitalized stay in the hospital for fewer than 5 days (median ~2-2.5 days per literature).

Yes, all patients included had blood withdrawn on days 1, 3, and 5, but they were not necessarily inpatients (only 51% admitted to hospital). We asked outpatients to come back on day 3 and 5 to the ED. Therefore we could generate complete datasets also for those patients not (anymore) in need of hospitalization or IV antibiotic. Of 339 randomized patients in the RCT, only two withdrew their consent, they were excluded from analyses. The rest of the patients had blood sampled on day 1, 3, 5. However, in this case “complete data sets” also refers to sufficient quantity of remaining stored plasma volume for this biomarker analysis. Therefore, only 175 datasets with enough stored blood plasma for copeptin and pro-ADM analysis were included in this analysis. Thus, we see no selection bias here. As we did not include this as explanation into the manuscript so far, we were happy to have changed the lines 227-232 to:

“The study population comprised 175 febrile LRTI pediatric patients (age: 1 month-18 years) for whom a sufficient quantity of blood plasma was in storage with febrile LRTI of whom a complete set of 3 consecutive blood samples were available for copeptin and MR-proADM measurements analysis for on day 1, 3, and 5 after study inclusion. of clinical presentation. Blood sampling was performed in-house while patients were hospitalized or in the emergency department, to where the outpatients and the discharged patients were asked to return on day 3 and 5.”

2. Line 215 – there are important data that are buried in Table S1. This table is labeled as a list of variables or data dictionary, when in fact the last column is important data to describe the study population and outcomes. This data should be in the main body of the paper.

Results: We inserted this data into Table 1 and deleted S1 Table as both table overlapped in content. 

3. Line 224 - Less than half the children in this study had pneumonia. The pathophysiology and microbiology of different lower respiratory tract infections are different, and therefore I’m concerned that this is a highly heterogeneous population (with a relatively small n, in addition). Children with bronchiolitis shouldn’t be requiring antibiotics to begin with and therefore it is not appropriate to include them with children with pneumonia in a primary outcome that includes IV antibiotic use.

Results: We refer to the third general reply to the major issues of this review further up. 1. Primary outcome was biomarkers` kinetics, secondary outcome were associations with other factors that could impact biomarkers` course over time. IV antibiotics was one of the few that showed a relevant association, not more, it should not be primary outcome. 2. We shy the differentiation of LRTI especially in young children, because case definitions are not very clear (e.g. CXR with highly variable interpretability are included for pneumonia), pathophysiology may overlap and antibiotic prescription is driven by fever. We rather would like to divide the cohort into those in need of antibiotics and those without, regardless of clinical diagnosis. Therefore we wanted to explore biomarkers` kinetics and associated factors in all febrile LRTI. 

For the sake of completeness diagnosis stratification was included into primary model, but showed no significant association.

4. Lines 246-252 – this section needs to be edited to explain the results in prose rather than relying on statistical parameters. The meaning of these results is lost in statistical jargon.

Results: The NLME base model description was moved to S1 Appendix. The results are presented now in a more accessible fashion and focused on the end results rather than including the description of the model.

5. Lines 264-266 – same as above – rather than talking about ‘baseline parameter’ and ‘slope parameter,’ this would be easier to read and interpret if the text read: “the following variables were statistically associated with initial levels of MR-proADM…ICU admission and microbiology results were associated with the change in MR-proADM from baseline to Day 5.” (if I’m interpreting the statistical jargon appropriately)

Results: We agree on this and have changed the entire result section and the table to become more accessible. It now reads:

“During forward covariate model building, antibiotic administration route, general complication, microbiology results and presence of fever had a significant influence on MR-proADM concentrations on day 1 (p<0.001). Admission to ICU and microbiology results were associated with the steepness of MR-proADM decrease. During the backward process, association with fever became not significant and was thus excluded. Details on base and final model can be seen in S1 Appendix. In the final model, intravenous administration of antibiotics influenced MR-proADM concentration and was significantly associated with a higher MR-proADM concentration on day 1. Further, ICU admission was significantly associated with a flatter MR-proADM concentration decrease from day 1 to day 5. In contrast, positive blood cultures, which were grouped together with negative NPA or no growth in blood culture because of the same effect when building the model, led to a significant decrease of MR-proADM (Table 1).”

 6. Table 2 and 3 – it would be helpful to have footnotes explaining the parameters in this table. What is meant by ‘limit?’

Results: The Tables have been modified and shifted in parts to the S1 Appendix, leaving a simplified single table in the main text body. Limit means the biomarker concentration at the end of the study. “Limit” was exchanged by “Copeptin/MR-proADM on day 5” to clarify this in Table 1 and 2 in S1 Appendix.

7. Line 302 – predict is the wrong term here. The biomarker is not doing the predicting – I would stick with MR-proADM discriminated those children receiving IV antibiotics vs those who did not. If less than half of the children in this study had pneumonia and hospitalized children with pneumonia generally warrant antibiotics, aren’t these results really a proxy for discriminating pneumonia from no pneumonia? There should be an examination of the interaction of diagnosis and IV antibiotics in these analyses.

Results: We do not want to include the interaction between diagnosis and IV antibiotics here, because this is planned to be part of a separate article that includes much more interactions not related to copeptin and pro-ADM and which be outside the scope of this present article. In the new version of this article we would keep kinetics as primary outcome and the factors impacting kinetics as secondary outcomes. For clarity, the antibiotic treatment in the ProPAED interventional group was PCT guided regardless of diagnosis, so only IV administration could serve as potential marker of severity.

8. Figure 2 – requires labels

Results: Figure 2 and the corresponding S6 Table were deleted due to the rather vague outcome parameter antibiotic administration route.

DISCUSSION

1. Line 333 – the discussion states that decreases were not as steep for CRP and PCT, yet the declines observed in Fig S1 do suggest that there were in fact declines in CRP and PCT. If this statement is going to be made, I would recommend some presentation of data in the Results section. In fact, I think that this manuscript would benefit from being reframed as biomarker kinetics in general, including CRP, PCT, copeptin and proADM, as that is really what this manuscript is about – not just copeptin and proADM.

Discussion: We did reframe this article as a biomarker article, focusing on kinetics and factors impacting kinetics. We excluded PCT and CRP from this version. We are planning a further article evaluating additional factors that go beyond the scope of this article. Further, PCT and CRP values have partly been published already, so we choose to exclude them.

2. Line 385 – include some discussion of Florin TA et al. Clin Infect Dis 2020 Aug 6; ciaa1138

Discussion: Included in lines 471.

3. Line 400-401 – just because the correlations were moderate here (some were not very good at all), one cannot conclude that combinations of biomarkers may be helpful. It would have been helpful if there was statistical analysis that focused on the outcomes of interest with combinations of biomarkers to be able to make this claim rather than simple correlations.

Discussion: We agree on this and have tempered our conclusions in line 528-533.

We are grateful to the staff and editors at PLOS ONE for the work they have done on our manuscript. This process helped to improve the quality of the manuscript. Please feel free to contact us anytime if further questions should arise.

Kind regards,

Philipp Baumann Jan Bonhoeffer

---

## [Decision Letter · Decision Letter 1]

14 Jan 2022

PONE-D-20-08646R1The kinetic profiles of copeptin and mid regional proadrenomedullin (MR-proADM) in pediatric lower respiratory tract infectionsPLOS ONE

Dear Dr. Baumann,

Thank you for submitting your manuscript to PLOS ONE. After careful consideration, we feel that it has merit but does not fully meet PLOS ONE’s publication criteria as it currently stands. Therefore, we invite you to submit a revised version of the manuscript that addresses the points raised during the review process.

Reviewers #1 and #2 think that their comments have been adequately responded. Your revision based upon the thid reviewer's suggestions also seem to be satisfactory. However, reviewer #1 additionally raised some minor issues that need to be addressed. Let us go one more round.

We look forward to receiving your revised manuscript.

Kind regards,

Yu Ru Kou, PhD

Academic Editor

PLOS ONE

Journal Requirements:

Reviewers' comments:

Reviewer's Responses to Questions

**Comments to the Author**

1. If the authors have adequately addressed your comments raised in a previous round of review and you feel that this manuscript is now acceptable for publication, you may indicate that here to bypass the “Comments to the Author” section, enter your conflict of interest statement in the “Confidential to Editor” section, and submit your "Accept" recommendation.

Reviewer #1: All comments have been addressed

Reviewer #2: All comments have been addressed

2. Is the manuscript technically sound, and do the data support the conclusions?

Reviewer #1: (No Response)

Reviewer #2: Yes

3. Has the statistical analysis been performed appropriately and rigorously? 

Reviewer #1: (No Response)

Reviewer #2: Yes

4. Have the authors made all data underlying the findings in their manuscript fully available?

Reviewer #1: (No Response)

Reviewer #2: Yes

5. Is the manuscript presented in an intelligible fashion and written in standard English?

Reviewer #1: (No Response)

Reviewer #2: Yes

6. Review Comments to the Author

Reviewer #1: Please abbreviate the 95% confidence intervals as 95%CI (not CI95%) in the abstract and everywhere else in the text.

Please replace all mentions of 'multivariate' analysis with 'multivariable' - all analyses here focused on one outcome at a time, not multiple outcomes in the same model - they are therefore multivariable NOT multivariate.

Please add 95% confidence intervals to the estimates of effect listed in Table 2 and wherever else they are cited in the text.

Reviewer #2: (No Response)

7. PLOS authors have the option to publish the peer review history of their article (what does this mean?). If published, this will include your full peer review and any attached files.

Reviewer #1: No

Reviewer #2: **Yes: **Samuel Uwaezuoke

---

## [Author Response · Author response to Decision Letter 1]

30 Jan 2022

Revised Manuscript PONE-D-20-08646R1: The kinetic profiles of copeptin and mid regional proadrenomedullin (MR-proADM) in pediatric lower respiratory tract infections

Dear Yu Ru Kou,

thank you very much for considering the publication of our manuscript. Here we answer point-by-point to the aspects raised by reviewer #1. Please see the corresponding corrections in the uploaded manuscript which we think is now suitable for publication. All line numbers in the revised manuscript correspond to the track-change version.

Reviewer #1: Please abbreviate the 95% confidence intervals as 95%CI (not CI95%) in the abstract and everywhere else in the text.

Authors` response: “CI95%” has been replaced by “95%CI” in Manuscript in lines 42, 43, and 46, and also in S1 Appendix.

Please replace all mentions of 'multivariate' analysis with 'multivariable' - all analyses here focused on one outcome at a time, not multiple outcomes in the same model - they are therefore multivariable NOT multivariate.

Authors` response: “Multivariate” has been replaced by “multivariable” in Manuscript in lines 197/198 (Table 2 caption) and in Table 2 itself. 

Please add 95% confidence intervals to the estimates of effect listed in Table 2 and wherever else they are cited in the text.

Authors` response: 95%CIs have been inserted into Table 2 as requested.

We are grateful to the reviewer and editors at PLOS ONE for the work they have done on our manuscript. Please feel free to contact us anytime if further questions should arise.

Kind regards,

Philipp Baumann

---

## [Editor Report · Decision Letter 2]

9 Feb 2022

The kinetic profiles of copeptin and mid regional proadrenomedullin (MR-proADM) in pediatric lower respiratory tract infections

PONE-D-20-08646R2

Dear Dr. Baumann,

We’re pleased to inform you that your manuscript has been judged scientifically suitable for publication and will be formally accepted for publication once it meets all outstanding technical requirements.

Kind regards,

Yu Ru Kou, PhD

Academic Editor

PLOS ONE
---

## [Editor Report · Acceptance letter]

1 Mar 2022

PONE-D-20-08646R2 

The kinetic profiles of copeptin and mid regional proadrenomedullin (MR-proADM) in pediatric lower respiratory tract infections 

Dear Dr. Baumann:

I'm pleased to inform you that your manuscript has been deemed suitable for publication in PLOS ONE. Congratulations! Your manuscript is now with our production department. 

Kind regards, 

on behalf of

Dr. Yu Ru Kou 

Academic Editor

PLOS ONE